# HSA: Head-wise Sparse Attention for Efficient and Accurate Long-context Inference

## Abstract

Transformer architectures have become the foundation of large language models (LLMs), excelling at sequential modeling via the self-attention mechanism. However, the quadratic computational complexity and linear KV cache growth of self-attention limit scalability in long-context scenarios. Sparse attention mechanisms, especially sliding window attention (SWA), help reduce these costs but inevitably constrain access to global context, which can degrade performance in tasks requiring long-range dependencies. While hybrid architectures that alternate between full-attention and SWA layers help mitigate this issue, their layer-wise sparsity pattern introduces a 'weakest-link' effect in which global context is inevitably lost in sparse layers, and the resulting degradation becomes more severe as the proportion of such layers increases. In this work, we introduce Head-wise Sparse Attention (HSA), a hybrid architecture that applies sparsity at the KV-head level. Unlike layer-wise sparse designs that impose a uniform sparsity pattern across all heads in a layer, HSA introduces sparsity at the KV-head level: a subset of heads is retained with full attention to preserve long-range dependencies, while the rest are converted to SWA for efficiency. This head-wise design ensures that every layer maintains global context through at least one full-attention KV head, while simultaneously reducing computation and KV-cache requirements. To decide which heads should remain global, we introduce a discrepancy-based post-training selection strategy that preserves those essential for capturing global context while converting the rest to sparse form. We then continue training to adapt the model to the new KV-head sparsity pattern. Extensive experiments on both public and in-house benchmarks show that HSA consistently outperforms prior layer-wise sparse designs, with the advantages being especially significant in long-context scenarios, while maintaining efficiency.

## 1 Introduction

Transformer architectures (Vaswani et al., 2017) have emerged as a cornerstone of large language models (LLMs), demonstrating remarkable versatility across a wide range of tasks. At the heart of this architecture is the self-attention mechanism, which excels at sequential modeling by capturing long-range dependencies and rich contextual relationships. However, the quadratic computational complexity $\mathcal{O}(N^2)$ of self-attention incurs substantial latency for long-context modeling. Moreover, in autoregressive LLM inference, where a prefill phase is followed by a decode phase, efficiency is achieved by caching key–value (KV) pairs from previous tokens. This KV cache grows linearly with sequence length $\mathcal{O}(N)$, further limiting scalability during inference. These limitations are particularly significant in reasoning-intensive tasks (Zelikman et al., 2022), which require referencing earlier information across multiple reasoning steps, and in multi-turn autonomous agent applications (Park et al., 2023) that must maintain long interaction histories.

To mitigate these issues, a straightforward approach is to exploit the inherent sparsity in attention patterns (Beltagy et al., 2020), thereby reducing the token-to-token computations in self-attention. By restricting each query to attend only a subset of keys and values—such as through fixed windows (Beltagy et al., 2020), dilated patterns (Beltagy et al., 2020), or content-based sparsity (Yuan et al., 2025)—both the computational cost and the KV-cache can be significantly reduced without severely degrading model performance. A representative example is sliding-window attention (SWA) (Child et al., 2019), where each token attends only to $w$ neighboring keys and values, reduc-

ing the computational complexity to $\mathcal{O}(Nw)$ and the KV-cache requirement to $\mathcal{O}(w)$. For instance, Mistral-7B (Jiang et al., 2023) adopts SWA with a fixed window size of 4096 to support longer sequences at modest additional cost. However, this locality constraint inevitably causes performance degradation on tasks that rely on long-range reasoning or cross-segment dependencies.

To address these limitations, recent work has explored hybrid architectures that alternate between full-attention layers and SWA layers, thereby providing periodic access to global context while retaining some of the efficiency benefits of sparsity. Notable examples include Gemma 2 (Team et al., 2024) and GPT-OSS (Agarwal et al., 2025), which interleave full-attention and sliding-window layers to balance accuracy and efficiency. However, this layer-wise sparsity pattern suffers from a "weakest-link" effect, as sparse layers inherently lack access to global context until the next full-attention layer. As the sparsity ratio increases, these local limitations accumulate, leading to substantial degradation in long-context performance. For instance, under high sparsity settings, layer-wise SWA results in severe performance drops (see Figure 2 and Table 2).

In this work, we propose Head-wise Sparse Attention (HSA), a hybrid architecture that applies sliding-window attention (SWA) at the KV-head level. Unlike layer-wise sparse designs that impose a uniform pattern across all heads within a layer, HSA assigns different patterns to different KV heads, ensuring that each layer retains at least one head with full attention to preserve long-range context, while the others adopt SWA for efficiency. This design is motivated by our empirical observation shown in Figure 1, which reveals that attention patterns vary across heads, with many focusing predominantly on local regions rather than global context. By adopting this finer-grained sparsification, HSA retains essential long-range dependencies while benefiting from the computational and KV-cache advantages of sparse attention. Specifically, we adapt an existing pre-trained model into a head-wise sparse architecture at the end of training, thereby avoiding the prohibitive cost of training from scratch. A key step in this process is identifying which KV heads can be sparsified without severely impairing global context modeling. The challenge lies in distinguishing locally focused KV heads, which can be replaced with SWA for efficiency, from globally oriented heads that should remain dense to preserve long-range dependencies. To address this, we introduce a simple yet effective discrepancy-based selection strategy. For each KV head in a pre-trained model, we measure the change in its attention output when that head is replaced with SWA. KV heads showing large discrepancies are considered critical for capturing global context and are retained with full attention, while those with small discrepancies are converted to SWA for efficiency. Finally, we perform training on the sparsified model to adapt it to the new sparsity pattern and enhance performance.

Our contributions can be summarized as follows:

- We propose HSA, a simple yet effective hybrid sparse attention framework that combines full attention and SWA at the KV-head level. Unlike prior layer-wise designs, HSA retains full attention for one or more heads while the remaining heads adopt sparse attention, ensuring that each layer preserves global context while reducing computation and KV-cache requirements.

- We introduce a simple, gradient-free criterion to determine which heads retain full attention and which adopt sparse attention, enabling efficient conversion of pre-trained models with only lightweight continued training instead of costly retraining from scratch.

- Extensive experiments on multiple large-scale MoE models demonstrate that HSA consistently outperforms strong layer-wise sparse baselines, achieving notable improvements on long-context benchmarks while maintaining strong performance on short-context tasks.

## 2 RELATED WORK

### 2.1 STATIC SPARSE ATTENTION

Static sparse attention refers to attention mechanisms where the sparsity pattern is fixed in advance rather than adapted dynamically for each input. These methods are simple to implement, computationally efficient, and hardware-friendly, while reducing KV-cache usage by restricting interactions to predetermined subsets of positions. Representative works (Child et al., 2019; Tay et al., 2020; Ainslie et al., 2020; Beltagy et al., 2020; Zaheer et al., 2020; Fu et al., 2025; Xiao et al., 2024; Gu et al., 2025) adopt hybrid static local/global patterns to lower compute and memory while preserving long-range dependencies. For example, Longformer (Beltagy et al., 2020) combines sliding-window

local attention with a small number of global tokens, while BigBird (Zaheer et al., 2020) augments local windows with random and global connections to balance sparsity and connectivity. Beyond global tokens, StreamingLLM (Xiao et al., 2024) highlights the role of attention sinks, typically the first few tokens in a sequence that consistently attract disproportionate attention across segments. Experiments show that removing these sink tokens during inference leads to noticeable performance drops, underscoring their importance in maintaining stable attention under sliding-window sparsity. More recently, DuoAttention (Xiao et al., 2025) introduces a head-wise hybrid design: some heads, called Retrieval Heads, maintain full attention and complete KV caches, while others, called Streaming Heads, operate with constant-length caches to reduce memory and latency in long-context inference. Meanwhile, Delta Attention (Willette et al., 2025) shows that static sparse methods often suffer from a distributional shift and proposes a lightweight correction mechanism that restores much of the lost accuracy. Compared with these approaches, HSA adopts a simple sliding-window mechanism with sink tokens and applies sparsity at the KV-head level through a discrepancy-driven selection process after pre-training. This enables existing pre-trained models to be adapted into hybrid architectures in a lightweight manner, while ensuring that each layer retains at least one global KV head to preserve long-range dependencies.

## 2.2 DYNAMIC SPARSE ATTENTION

Dynamic sparse attention methods adapt the sparsity pattern based on the input or inference context instead of relying on a fixed mask. Their goal is to preserve global information while flexibly accommodating varying contextual demands. Recent advances, such as Native Sparse Attention (NSA) (Yuan et al., 2025), follow this direction by dynamically selecting attention connections according to content relevance. Other techniques explore adaptive token selection (Zhang et al., 2025; Kitaev et al., 2020; Lu et al., 2025), routing (Roy et al., 2021; Jiang et al., 2024), or pruning policies (Wang et al., 2021; Zhang et al., 2023; Mu et al., 2023; Ge et al., 2024) that tailor attention spans to specific inputs. While these approaches offer greater adaptability than static patterns, their dynamic nature poses practical challenges. Many methods (Yuan et al., 2025; Jiang et al., 2024) cannot reduce KV cache size, since each query may attend to a different subset of keys, requiring storage of the full cache. Even when partial KV cache pruning is applied (Zhang et al., 2023; Ge et al., 2024), it risks discarding information that may later be needed, leading to performance degradation. Moreover, dynamic mechanisms often introduce additional runtime and implementation overhead, which can complicate efficient deployment at scale. In contrast, our method remains within the static sparse paradigm but introduces flexibility by assigning different sparsity patterns to different KV heads. This head-wise formulation preserves the simplicity and efficiency of static designs, avoids the overhead of dynamic mechanisms, and enhances context retention by ensuring that each layer maintains at least one global KV head.

## 3 APPROACH

In this section, we first review the formulation of multi-head attention along with its computational cost and KV-cache usage. We then present HSA, describing its KV-head–wise sparsification framework, discrepancy-based head selection strategy, and theoretical efficiency analysis.

## 3.1 PRELIMINARY: MULTI-HEAD ATTENTION

For notational simplicity, we present the formulation in the standard multi-head attention (MHA) setting and omit the layer index for clarity. The extension to grouped-query attention (GQA) (Ainslie et al., 2023) is analogous, except that multiple query heads share the same set of key–value heads. Given an input $\mathbf{X} \in \mathbb{R}^{N \times D}$ to a multi-head attention layer (Vaswani et al., 2017), where $N$ is the sequence length and $D$ the model dimension, the input is projected into query, key, and value representations using three learnable matrices $\mathbf{W}_Q^h, \mathbf{W}_K^h, \mathbf{W}_V^h \in \mathbb{R}^{D \times d}$ for the $h$-th head:

$$\mathbf{Q}^h = \mathbf{X}\mathbf{W}_Q^h, \quad \mathbf{K}^h = \mathbf{X}\mathbf{W}_K^h, \quad \mathbf{V}^h = \mathbf{X}\mathbf{W}_V^h, \tag{1}$$

where $d$ denotes the head dimension. For each head, the attention output $\mathbf{O}^h$ is then computed as a weighted sum of the values:

$$\mathbf{O}^h = \text{Attention}(\mathbf{Q}^h, \mathbf{K}^h, \mathbf{V}^h) = \text{Softmax}(\frac{\mathbf{Q}^h\mathbf{K}^{h\top}}{\sqrt{d}})\mathbf{V}^h. \tag{2}$$

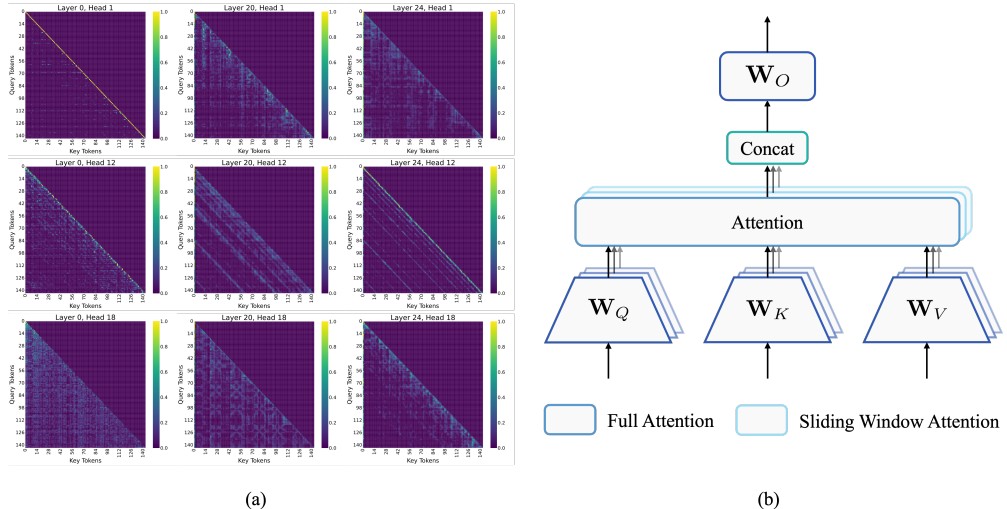

(a)                (b)

Figure 1: (a) Attention maps from different layers and heads of our in-house MoE-2.5B/50B, showing that some heads capture global dependencies while many focus mainly on local neighborhoods. (b) HSA assigns different attention patterns to individual heads within the same layer. A subset of heads operate with full attention to preserve global context, while others adopt SWA to reduce computational cost and KV cache storage.

Then, the output of each head is concatenated and projected back to the model dimension through a learnable matrix $\mathbf{W}_O \in \mathbb{R}^{Hd \times D}$, where $H$ is the number of heads:

$$\mathbf{Z} = \mathrm{Concat}(\mathbf{O}^1, \mathbf{O}^2, \ldots, \mathbf{O}^H)\mathbf{W}_O. \tag{3}$$

With $N$ tokens, the computational complexity of standard multi-head attention is $\mathcal{O}(N^2 dH)$, primarily due to the query–key dot products. During autoregressive inference, the key–value (KV) cache must be maintained to enable efficient decoding. In particular, once the prefill phase is completed, all keys and values from the $N$ input tokens need to be stored and reused for subsequent generation steps, incurring a memory cost of $\mathcal{O}(NdH)$ across all heads. As the sequence length $N$ increases, both computation and memory become prohibitive. The quadratic computational complexity $\mathcal{O}(N^2 dH)$ rapidly dominates runtime, making prefill latency grow superlinearly with $N$. At the same time, the KV cache grows linearly as $\mathcal{O}(NdH)$, which leads to substantial memory overhead during decoding.

### 3.2 Head-wise Sparse Attention

To reduce both the quadratic computational complexity of self-attention and the linear KV cache growth, we propose head-wise sparse attention (HSA), a hybrid attention mechanism that introduces sparsity at the granularity of individual heads, as shown in Figure 1. Unlike layer-wise sparse designs such as Gemma 2 (Team et al., 2024) and GPT-OSS (Agarwal et al., 2025), which impose a uniform sparsity pattern across all heads within a layer, HSA introduces sparsity at the finer granularity of individual KV heads. This distinction is crucial: sparsity is applied only to the KV heads, while the queries remain dense, ensuring that efficiency gains lead to reduced computation and smaller KV-cache requirements. Formally, we define a sparsity ratio $\rho \in [0, 1]$ to control the proportion of KV heads converted to sparse attention. A fraction $\rho$ of the KV heads are replaced with sliding-window attention (SWA) equipped with an attention-sink mechanism following StreamingLLM (Xiao et al., 2024), whose purpose is to retain as much of the original attention score distribution as possible despite the locality constraint, thereby mitigating information loss. The remaining $1 - \rho$ fraction are kept as full attention to preserve long-range dependencies. To further maintain global information, HSA requires that each layer retain at least one full-attention KV head. This KV-head–wise design offers finer granularity than layer-wise sparsity, enabling a more balanced trade-off between efficiency and context preservation, while simultaneously reducing computational cost and KV-cache requirements without fully discarding global context.

### 3.3 Discrepancy-based KV-head Selection

In practice, HSA is constructed from an existing pre-trained model rather than trained from scratch, as the latter would be prohibitively expensive for large-scale LLMs. Given such a model, the key step is to determine which KV heads should be sparsified. Intuitively, not all KV heads contribute equally to modeling long-range dependencies: some specialize in capturing global context, while others primarily focus on local neighborhoods. Replacing a globally oriented KV head with a sparse alternative is likely to cause a substantial deviation in the model output, whereas substituting a locally focused KV head tends to produce comparatively smaller deviations. Motivated by this intuition, we adopt a discrepancy-based KV-head selection strategy that quantifies the output difference introduced by sparsification. For a given layer, we replace the $h$-th KV head with SWA and measure the resulting output discrepancy $\Delta^h$ by

$$\Delta^h = \left\| \mathrm{Attention}(\mathbf{Q}^h, \mathbf{K}^h, \mathbf{V}^h)\mathbf{W}_O^h - \mathrm{SWA}(\mathbf{Q}^h, \mathbf{K}^h, \mathbf{V}^h)\mathbf{W}_O^h \right\|, \tag{4}$$

where $\mathrm{SWA}(\cdot, \cdot, \cdot)$ denotes the sliding window attention operator with window size of $w$. The discrepancy $\Delta^h$ is computed on a small calibration dataset to quantify the sensitivity of each head to sparsification. Heads with large $\Delta^h$ values are retained as full attention to preserve global context, whereas those with small values are replaced by SWA. Given a sparsity ratio $\rho$, the overall algorithm for discrepancy-based KV-head selection is summarized in Algorithm 1. We then continue training the resulting sparsified model to adapt the model to the new KV-head configuration and improve performance under the modified sparsity pattern.

---

**Algorithm 1:** Discrepancy-based KV-head Selection

---

**Input:** Input $\mathbf{X} \in \mathbb{R}^{N \times D}$, projection weights $\mathbf{W}_Q^h, \mathbf{W}_K^h, \mathbf{W}_V^h \in \mathbb{R}^{D \times d}$, sparsity ratio $\rho$.
**Output:** Index set $\mathcal{H}_{\mathrm{swa}} \subseteq \{1, \ldots, H\}$ with $|\mathcal{H}_{\mathrm{swa}}| = \rho H$.
Initialize an empty list $S$;
**for** $h \in \{1, 2, \ldots, H\}$ **do**
 Compute $\mathbf{Q}^h$, $\mathbf{K}^h$, and $\mathbf{V}^h$ using Eq. (1);
 Compute the discrepancy score $\Delta^h$ using Eq. (4) and record the pair $(h, \Delta^h)$ in $S$;

Select the $\rho H$ heads with the smallest $\Delta^h$ values in $S$ and denote their indices as $\mathcal{H}_{\mathrm{swa}}$;
**return** $\mathcal{H}_{\mathrm{swa}}$;

---

### 3.4 Efficiency Discussion

**Selection efficiency.** Our discrepancy-based KV-head selection is highly efficient because it operates in a gradient-free manner, eliminating the need for backward propagation. Instead of re-training or fine-tuning to determine head importance, we simply measure the output discrepancy on a small calibration set, which requires only forward passes. This drastically reduces the computational overhead compared to gradient-based pruning or training-time head reallocation. In practice, the selection step adds negligible cost relative to model pre-training, making it scalable even for very large LLMs.

**Computation and KV-cache reduction.** In HSA, we replace a proportion $\rho$ of the KV heads with SWA using a window size $w \ll N$. We consider causal attention under the convention that each query can attend to all preceding tokens including itself. For a sequence of length $N$, the $t$-th query attends to exactly $t$ keys. Summing over all positions yields the total number of query–key pairs per head in full attention:

$$S_{\mathrm{full}} = \sum_{t=1}^{N} t = \frac{N(N+1)}{2}. \tag{5}$$

For sliding-window attention with window size $w$, the $t$-th query attends to $\min(t, w)$ keys. Therefore, the total number of pairs becomes

$$S_{\mathrm{SWA}} = \sum_{t=1}^{N} \min(t, w) = \underbrace{\sum_{t=1}^{w} t}_{\text{growing window}} + \underbrace{\sum_{t=w+1}^{N} w}_{\text{fixed window}} = \frac{w(w+1)}{2} + (N-w)w. \tag{6}$$

This simplifies to

$$S_{\text{SWA}} = wN - \frac{w(w-1)}{2}. \tag{7}$$

The relative cost of SWA compared to full attention is

$$\frac{S_{\text{SWA}}}{S_{\text{full}}} = \frac{(2\beta - \beta^2)N + \beta}{N + 1}, \text{where } \beta = \frac{w}{N}. \tag{8}$$

Accordingly, the overall attention computation cost in HSA is $2dH\left[(1 - \rho)\,S_{\text{full}} + \rho S_{\text{SWA}}\right]$ while the KV-cache storage cost becomes $\mathcal{O}(Nd(1 - \rho)H + wd\rho H)$. Relative to full causal attention, the computation speedup ratio of HSA is

$$\frac{S_{\text{full}}}{(1 - \rho)S_{\text{full}} + \rho S_{\text{SWA}}} = \frac{1}{(1 - \rho) + \rho \cdot \frac{(2\beta - \beta^2)N + \beta}{N + 1}} \text{ where } \beta = \frac{w}{N}. \tag{9}$$

For KV-cache storage, the compression ratio is $\frac{1}{1 - \rho + \rho\beta}$. As a concrete example, with $\rho = 0.75$, $w = 4K$, and $N = 32K$ ($\beta = 1/8$), HSA achieves a computation speedup of $\sim 2.35\times$ and a KV-cache compression of $\sim 2.91\times$, substantially reducing both FLOPs and memory costs.

## 4 EXPERIMENTS

**Experimental settings.** We evaluate HSA on two in-house MoE models, MoE-680M/13.6B and MoE-2.5B/50B, trained on proprietary in-house datasets. Additional experiments on the open-source dense model OLMo 2 7B (OLMo et al., 2024) are presented in Section C of the appendix. MoE-680M/13.6B is first pre-trained on 400B tokens with a maximum sequence length of 8K, after which HSA is applied and the model is further adapted through continued training on 100B tokens with an extended sequence length of 32K. MoE-2.5B/50B follows the same pipeline at a larger scale, with 500B tokens for pre-training and 200B tokens for continued training. Unless otherwise stated, the sliding-window size for SWA heads is set to 4K, and sparsity ratios of $\rho = 0.5$ and $\rho = 0.75$ are explored. For head selection, we randomly sample 512 instances from the 32K sequence-length training set, consisting of 256 English and 256 Chinese samples. We compare HSA against layer-wise SWA with four attention sinks (Xiao et al., 2024), using identical training configurations across all methods. In addition, the first and last layers are kept as full attention in all cases. For short-context evaluation, we report performance on widely used open-source reasoning benchmarks under few-shot settings, including ARC-Challenge (Clark et al., 2018), BBH (Suzgun et al., 2023), HellaSwag (Zellers et al., 2019), MMLU (Hendrycks et al., 2021), MMLU-Pro (Wang et al., 2024), C-Eval (Huang et al., 2023), and WinoGrande (Sakaguchi et al., 2021). For long-context evaluation, we assess performance on LongBench (Bai et al., 2024) and RULER (Hsieh et al., 2024). We further evaluate on internal long-context benchmarks, including the Needle-in-a-Haystack test and diverse retrieval, reasoning, and comprehension tasks up to 32K tokens.

Table 1: Performance of MoE-2.5B/50B on short-context benchmarks across different methods.

| Method | ARC-c | BBH | HellaSwag | WinoGrande | MMLU | MMLU-Pro | C-Eval | Avg. |
|---|---|---|---|---|---|---|---|---|
| **Baseline** | 88.8 | 64.7 | 75.7 | 75.9 | 74.3 | 79.1 | 46.9 | 72.2 |
| **Layer-wise SWA** ($\rho = 0.5$) | 88.6 | 65.4 | **75.7** | **76.2** | **74.1** | 79.0 | **46.5** | 72.2 |
| **HSA** ($\rho = 0.5$) | **89.2** | **65.5** | 75.4 | 75.5 | 73.9 | **80.6** | **46.5** | **72.4** |
| **Layer-wise SWA** ($\rho = 0.75$) | 88.9 | 64.1 | 75.4 | 75.3 | 73.8 | 79.2 | 47.2 | 72.0 |
| **HSA** ($\rho = 0.75$) | **89.4** | **64.3** | **75.7** | **75.7** | **74.3** | **80.3** | **47.6** | **72.5** |

### 4.1 MAIN RESULTS

We present the results of MoE-2.5B/50B in Tables 1 and 2, as well as in Figure 2. More results of MoE-680M/13.6B can be found in Section B of the appendix. On short-context benchmarks, all methods perform comparably to the baseline, as the 4K window size is already sufficient to cover nearly the entire input. Notably, HSA consistently achieves slightly better results than layer-wise SWA across different sparsity ratios. For example, at $\rho = 0.75$, the average accuracy improves from

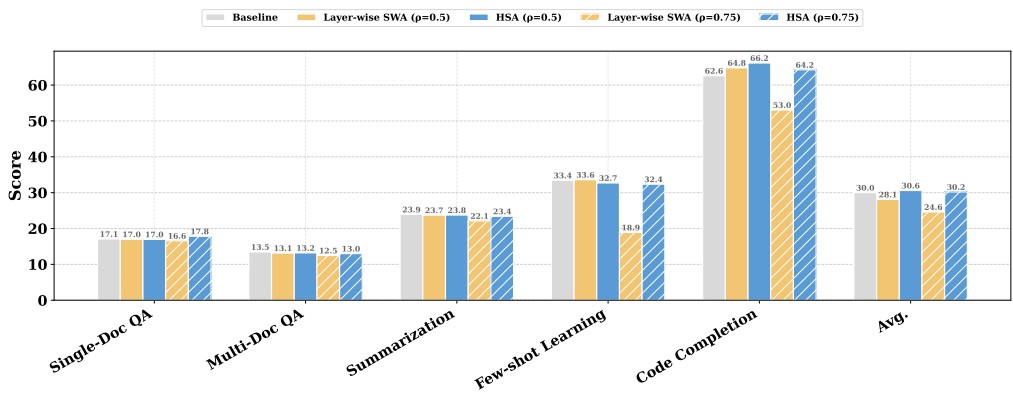

Figure 2: Performance of MoE-2.5B/50B on LongBench across different methods. Detailed results for individual subsets are provided in Section A of the appendix.

Table 2: Performance of MoE-2.5B/50B across different methods on RULER and in-house long-context benchmarks.

| Model | RULER | | | | | | In-house Evaluation | | |
| --- | --- | --- | --- | --- | --- | --- | --- | --- | --- |
| | 4K | 8K | 16K | 32K | 64K | 128K | NIAH | Others | Avg. |
| **Baseline** | 93.2 | 89.5 | 87.0 | 80.1 | 45.1 | 30.0 | 83.8 | 27.4 | 52.6 |
| **Layer-wise SWA** ($\rho = 0.5$) | **95.2** | **91.7** | 86.2 | 74.8 | 41.4 | 25.8 | **82.6** | 27.4 | 52.2 |
| **HSA** ($\rho = 0.5$) | 94.4 | 91.3 | **86.7** | **83.0** | **48.5** | **31.0** | 79.9 | **29.1** | **53.1** |
| **Layer-wise SWA** ($\rho = 0.75$) | 94.5 | 85.6 | 78.3 | 64.8 | 34.2 | 21.3 | 69.5 | 25.9 | 46.7 |
| **HSA** ($\rho = 0.75$) | **94.8** | **91.2** | **87.9** | **81.6** | **46.1** | **29.0** | **78.4** | **30.1** | **53.6** |

72.0% to 72.5%, even outperforms baseline by 0.3%. The advantages of HSA become far more pronounced on long-context benchmarks, where modeling dependencies beyond the local window is essential. In these settings, the limitations of layer-wise sparsity become apparent: once global context is dropped in a sparse layer, it cannot be preserved until a subsequent full-attention layer, and the resulting degradation compounds as sparsity increases. At a sparsity ratio of 0.75, for instance, layer-wise SWA suffers average performance drops of 5.4%, 15.3%, and 5.9% on Long-Bench, 32K RULER, and our in-house long-context benchmark, respectively. In contrast, HSA not only avoids such degradation but even outperforms the baseline, with gains of 0.2%, 1.5%, and 1.0% on the same benchmarks. Note that our models are trained with sequences up to 32K, so evaluations beyond this length correspond to extrapolation. Even under extrapolation settings, HSA delivers notable gains over layer-wise SWA, achieving an 11.9% score improvement on 64K RULER. These results highlight the effectiveness of HSA's KV-head–wise design. By ensuring that at least one global KV head is preserved in every layer, HSA maintains access to long-range information across the entire network, while selectively sparsifying locally focused heads for efficiency. This avoids the weakest-link effect observed in layer-wise designs and enables more consistent performance as sparsity increases. Overall, the findings demonstrate that HSA consistently provides advantages across sparsity levels, offering small but consistent improvements in short-context tasks and substantial gains in long-context settings, particularly under high sparsity.

## 4.2 FURTHER STUDIES

**Effect of keeping attention sinks.** Since HSA is built upon sliding-window attention, we further examine the role of attention sinks by comparing two variants: HSA (with sinks) and HSA w/o sink (sinks removed, where "w/o" denotes "without"). Both variants use the same sparsity ratios $\rho$, window sizes $w$, training schedules, and datasets, with the only difference being whether sink tokens are preserved. As shown in Table 3, retaining attention sinks consistently improves performance across benchmarks. For instance, on RULER 32K, HSA with attention sinks outperforms the variant without by 6%. This aligns with observations in StreamingLLM (Xiao et al., 2024), where the first few tokens serve as persistent "sinks" that stabilize attention across segments. Based on these results, we adopt HSA with attention sinks as the default in all subsequent experiments.

Table 3: Effect of retaining attention sinks. We report performance of MoE-2.5B/50B on RULER and in-house long-context benchmarks.

| Model | RULER | | | | | | In-house Evaluation | | |
|---|---|---|---|---|---|---|---|---|---|
| | 4K | 8K | 16K | 32K | 64K | 128K | NIAH | Others | Avg. |
| **Baseline** | 93.2 | 89.5 | 87.0 | 80.1 | 45.1 | 30.0 | 83.8 | 27.4 | 52.6 |
| **HSA w/o sink** ($\rho = 0.75$) | 94.0 | 90.2 | 87.8 | 75.6 | 41.8 | 26.4 | **78.4** | 28.3 | 51.9 |
| **HSA** ($\rho = 0.75$) | **94.8** | **91.2** | **87.9** | **81.6** | **46.1** | **29.0** | **78.4** | **30.1** | **53.6** |

**Effect of different window sizes.** We investigate how varying the sliding-window size $w \in \{1K, 2K, 4K\}$ impacts model performance. Experiments are conducted on MoE-680M/13.6B with a sparsity ratio of $\rho = 0.75$, and the results are reported in Table 4. As expected, smaller windows increase efficiency by reducing both computational cost and KV-cache usage, but they also restrict the receptive field of sparse heads, limiting the ability to capture long-range dependencies. Larger windows alleviate this issue by incorporating broader context, albeit at the expense of efficiency. In practice, shrinking the window size leads to a slight degradation on both short- and long-context benchmarks. Nevertheless, since HSA retains full-attention heads in every layer, the overall performance drop remains modest. Balancing these trade-offs, we adopt a window size of 4K as the default setting, which offers a favorable compromise between efficiency and accuracy across tasks.

Table 4: Effect of different window sizes. We report performance of MoE-680M/13.6B on short-context benchmarks and in-house long-context benchmarks.

| Model | Short-context benchmarks | | | | | | | | In-house Evaluation | | |
|---|---|---|---|---|---|---|---|---|---|---|---|
| | ARC-c | BBH | HellaSwag | WinoGrande | MMLU | MMLU-Pro | C-Eval | Avg. | NIAH | Others | Avg. |
| **Baseline** | 81.6 | 51.1 | 69.3 | 70.2 | 65.5 | 73.5 | 34.0 | 63.6 | 66.9 | 20.2 | 40.2 |
| $w = 1K$ | 80.9 | 49.1 | 69.6 | 68.1 | **65.8** | 73.6 | 33.8 | 63.0 | **58.2** | 18.7 | 36.2 |
| $w = 2K$ | **80.7** | **49.9** | 69.8 | 68.7 | 65.5 | **74.0** | 33.9 | **63.2** | 55.8 | **19.7** | 36.4 |
| $w = 4K$ | **80.7** | 49.4 | **69.8** | **69.5** | 65.6 | 73.3 | **34.0** | **63.2** | 58.1 | 19.4 | **36.8** |

**Effect of discrepancy-based KV-head selection.** We compare our proposed discrepancy-based KV-head selection (DBKS) against two attention-map–based variants. The first variant, AM-DBKS, selects heads solely based on the difference between the attention maps of full and sparse attention. The second, AMV-DBKS, extends this approach by also incorporating values, where head selection is guided by the discrepancy between attention-weighted outputs, *i.e.*, $\Delta^h = \left\| \text{Attention}(\mathbf{Q}^h, \mathbf{K}^h, \mathbf{V}^h) - \text{SWA}(\mathbf{Q}^h, \mathbf{K}^h, \mathbf{V}^h) \right\|$. In contrast, our DBKS directly measures the output-level discrepancy defined in Eq. (4). Experiments are conducted on MoE-680M/13.6B at a sparsity ratio of $\rho = 0.75$, and the results are shown in Table 5. As shown, AM-DBKS is inherently limited because it only compares attention distributions and neglects the role of values in forming the final representation. AMV-DBKS improves upon this by incorporating values into the selection process, which leads to much stronger performance, but it still overlooks the influence of the output projection weights $\mathbf{W}_O$. In contrast, our DBKS measures discrepancy directly at the output level, taking into account the joint effects of attention weights, value vectors, and output projection, which explains its consistently superior performance. For instance, DBKS surpasses AMV-DBKS by 4.5% on 32K RULER and 2.2% on our in-house long-context benchmarks.

Table 5: Effect of different KV-head selection strategies. We report performance of MoE-680M/13.6B on RULER and in-house long-context benchmarks.

| Model | RULER | | | | | | In-house Evaluation | | |
|---|---|---|---|---|---|---|---|---|---|
| | 4K | 8K | 16K | 32K | 64K | 128K | NIAH | Others | Avg. |
| **AM-DBKS** | 89.8 | 70.0 | 61.4 | 51.4 | 30.5 | 17.8 | 50.4 | 18.5 | 33.6 |
| **AMV-DBKS** | **90.6** | 78.1 | 70.5 | 57.4 | **33.0** | **20.0** | 53.5 | 18.6 | 34.6 |
| **DBKS** | 89.7 | **78.9** | **71.1** | **61.9** | 32.7 | 19.3 | **58.1** | **19.4** | **36.8** |

**Effect of different data sizes for KV-head selection.** We study how the number of samples influences KV-head selection by conducting experiments on MoE-680M/13.6B at a sparsity ratio of $\rho = 0.75$ with varying data sizes. As shown in Table 6, larger sample sets enable more accurate identification of informative heads and yield stronger downstream performance, as they provide a more reliable signal for selection. Among the tested settings, using 512 samples achieves the best overall performance, and we therefore adopt it as the default in our experiments.

Table 6: Effect of different data sizes for KV-head selection. Results are shown for MoE-680M/13.6B on RULER and our in-house long-context benchmarks.

| Model | RULER | | | | | | In-house Evaluation | | |
|---|---|---|---|---|---|---|---|---|---|
| | 4K | 8K | 16K | 32K | 64K | 128K | NIAH | Others | Avg. |
| **128** | **90.0** | 78.2 | **71.8** | 57.3 | **32.9** | 19.0 | 53.4 | **19.5** | 35.5 |
| **256** | 88.4 | 78.2 | 69.8 | 56.6 | 32.8 | 19.1 | 53.1 | 20.1 | 36.0 |
| **512** | 89.7 | **78.9** | 71.1 | **61.9** | 32.7 | **19.3** | **58.1** | 19.4 | **36.8** |

**Computational efficiency.** To evaluate the computational efficiency of HSA, we measure the forward and backward pass time of a stack of four attention modules on a GPU accelerator, using the Qwen3-8B (Team, 2025) configuration with 32 query heads, 8 KV heads, and a head dimension of 128. The sparsity ratio is fixed at $\rho = 0.75$, the window size at $w = 4096$, and the batch size at 1. We compare three settings: (i) Baseline, where all four modules use full attention; (ii) Layerwise SWA, where one module uses full attention and the remaining three use SWA; and (iii) HSA, where all four modules adopt HSA with $\rho = 0.75$. All attention computations are executed with the official FlashAttention kernel (Dao et al., 2022); in HSA, heads are dispatched to either dense FlashAttention or SWA FlashAttention through PyTorch (Ansel et al., 2024). As shown in Figure 3, HSA achieves speedups comparable to layer-wise SWA, confirming that head-wise sparsification introduces little additional overhead while still providing significant efficiency gains. More importantly, unlike layer-wise SWA, which suffers from accuracy degradation under high sparsity, HSA achieves markedly better performance across long-context benchmarks (see Figure 2 and Table 2). Furthermore, the efficiency benefit scales with context length: at a sequence length of 128K, HSA achieves speedups of $3.31\times$ and $3.28\times$ in forward and backward passes compared to full attention.

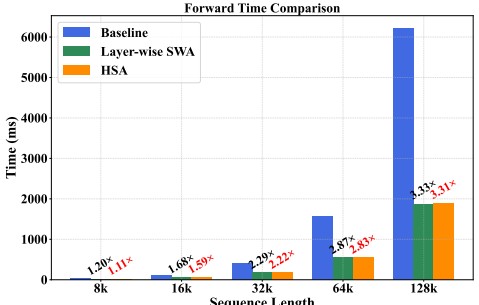 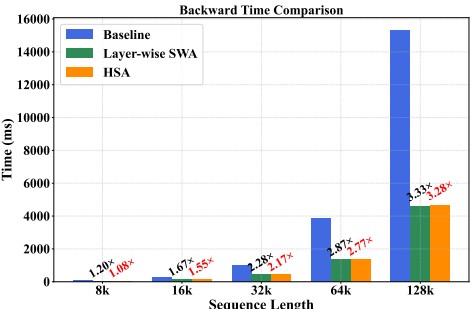

Figure 3: Forward and backward time comparison for four attention modules.

## 5 CONCLUSION

In this work, we have proposed HSA, a hybrid architecture that introduces sparsity at the KV-head level. By selectively converting locally focused heads into SWA while retaining globally oriented ones, HSA achieves a finer balance between efficiency and context preservation. Through discrepancy-based head selection and continued training, we have demonstrated that HSA can be seamlessly applied to pre-trained models, reducing computational cost and KV-cache requirements while largely preserving global context. Beyond its efficiency gains, HSA underscores the importance of head-level granularity in sparse attention design, offering a perspective that complements existing layer-wise sparse approaches. Extensive experiments on both public and in-house benchmarks confirm its effectiveness, showing consistent improvements on short-context tasks and substantial gains in long-context scenarios, particularly under high sparsity.

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

# Appendix

## USAGE OF LARGE LANGUAGE MODELS.

We employed ChatGPT to help refine and improve the presentation of this paper. Some figures were also initially produced using code generated by large language models.

## A   ADDITIONAL RESULTS OF MoE-2.5B/50B ON LONGBENCH

Table A reports detailed results of MoE-2.5B/50B on LongBench. Across most subsets, HSA surpasses layer-wise SWA, demonstrating stronger ability to preserve long-range dependencies under different sparsity ratios.

Table A: Performance comparisons of MoE-2.5B/50B on LongBench.

| Model | Single-Doc QA | | | | Multi-Doc QA | | | | Summarization | | | | Few-shot Learning | | | | Code Completion | | Avg. |
|---|---|---|---|---|---|---|---|---|---|---|---|---|---|---|---|---|---|---|---|
| | NQA | QQA | MFQA-en | MFQA-zh | HQA | 2WM | Mus | DuR | GvR | QMS | MNs | VCS | TRC | TQA | SSM | LSHT | LCC | RBP | |
| Baseline | 3.2 | 5.9 | 7.5 | 51.7 | 11.9 | 10.3 | 6.8 | 24.8 | 30.9 | 22.5 | 26.4 | 15.9 | 44.0 | 21.6 | 27.3 | 40.8 | 66.1 | 59.1 | 30.0 |
| Layer-wise SWA ($\rho = 0.5$) | 3.0 | 6.1 | **7.4** | **51.5** | **11.2** | **10.6** | 6.3 | 24.5 | 29.5 | **23.2** | **26.0** | 16.2 | 43.0 | 21.8 | **29.9** | **39.7** | 68.0 | 61.6 | 30.5 |
| HSA ($\rho = 0.5$) | **3.4** | **6.2** | 7.3 | 51.0 | 10.9 | 10.3 | **6.5** | **25.1** | **30.4** | 22.3 | 25.8 | **16.5** | **44.3** | **23.0** | 28.5 | 35.0 | **68.9** | **63.4** | **30.6** |
| Layer-wise SWA ($\rho = 0.75$) | 4.5 | 6.1 | 7.0 | 48.7 | 9.8 | **9.7** | 5.6 | 24.8 | 24.9 | 21.6 | 26.0 | 16.0 | 21.6 | 18.3 | 9.8 | 25.9 | 55.6 | 50.5 | 24.6 |
| HSA ($\rho = 0.75$) | **6.4** | **6.4** | **7.2** | **51.3** | **11.7** | 9.4 | **5.9** | **25.0** | **28.1** | **23.1** | **26.2** | **16.2** | **43.0** | **22.7** | **27.5** | **36.2** | **68.6** | **59.9** | **30.2** |

## B   MORE RESULTS OF MoE-680M/13.6B

We provide additional results for MoE-680M/13.6B in Tables B, C, and D. From the results, we observe that on short-context benchmarks, all methods perform comparably to the baseline. On long-context evaluations, however, HSA demonstrates clear advantages over layer-wise SWA, alleviating the compounding degradation of sparsity—for instance, on 16K RULER, HSA outperforms layer-wise SWA by 12.6%. While the absolute performance of HSA is slightly below the baseline due to the limited amount of continued training data at this scale, it still delivers significant gains over layer-wise SWA.

Table B: Performance of MoE-680M/13.6B on short-context benchmarks across different methods.

| Method | ARC-c | BBH | HellaSwag | WinoGrande | MMLU | MMLU-Pro | C-Eval | Avg. |
|---|---|---|---|---|---|---|---|---|
| **Baseline** | 81.6 | 51.1 | 69.3 | 70.2 | 65.5 | 73.5 | 34.0 | 63.6 |
| **Layer-wise SWA** ($\rho = 0.75$) | **81.6** | 49.1 | 69.7 | 69.1 | **65.9** | **74.7** | **34.5** | **63.5** |
| **HSA** ($\rho = 0.75$) | 80.7 | **49.4** | **69.8** | **69.5** | 65.6 | 73.3 | 34.0 | 63.2 |

Table C: Performance of MoE-680M/13.6B on RULER and internal long-context evaluation datasets across different methods.

| Model | RULER | | | | | | In-house Evaluation | | |
|---|---|---|---|---|---|---|---|---|---|
| | 4K | 8K | 16K | 32K | 64K | 128K | NIAH | Others | Avg. |
| **Baseline** | 89.9 | 84.2 | 77.0 | 66.1 | 37.3 | 21.5 | 66.9 | 20.2 | 40.2 |
| **Layer-wise SWA** ($\rho = 0.75$) | 88.8 | 68.7 | 58.5 | 59.1 | 31.6 | 18.5 | 55.4 | 18.3 | 35.0 |
| **HSA** ($\rho = 0.75$) | **89.7** | **78.9** | **71.1** | **61.9** | **32.7** | 19.3 | **58.1** | **19.4** | **36.8** |

Table D: Performance of MoE-680M/13.6B on LongBench across different methods.

| Model | Single-Doc QA | | | | Multi-Doc QA | | | | Summarization | | | | Few-shot Learning | | | | Code Completion | | Avg. |
|---|---|---|---|---|---|---|---|---|---|---|---|---|---|---|---|---|---|---|---|
| | NQA | QQA | MFQA-en | MFQA-zh | HQA | 2WM | Mus | DuR | GvR | QMS | MNs | VCS | TRC | TQA | SSM | LSHT | LCC | RBP | |
| Baseline | 2.6 | 6.2 | 7.1 | 47.1 | 9.8 | 9.4 | 5.5 | 24.5 | 26.4 | 23.0 | 27.1 | 15.1 | 39.0 | 22.5 | 32.9 | 37.9 | 55.6 | 51.1 | 27.5 |
| Layer-wise SWA ($\rho = 0.75$) | 2.7 | 6.1 | 6.9 | 46.3 | **9.2** | 8.8 | **5.8** | **24.8** | **28.0** | 19.9 | **26.2** | 14.3 | **38.8** | **23.2** | 33.0 | 31.3 | 58.6 | 52.8 | 27.4 |
| HSA ($\rho = 0.75$) | **4.6** | **6.4** | **7.1** | **49.1** | 8.5 | **9.2** | 5.3 | 24.7 | 28.5 | **22.1** | 25.6 | **15.2** | 38.3 | 23.0 | **33.8** | **32.8** | **60.5** | **53.7** | **28.1** |

# C   MORE RESULTS OF OLMO 2 7B

**Experimental settings.** In addition to our in-house MoE models, we evaluate HSA on the open-source OLMo 2 7B (OLMo et al., 2024). The model is pre-trained on 4T tokens and further adapted with an additional 50B tokens during the mid-training stage, following the official OLMo 2 protocol. Unlike our in-house models, both stages use a training sequence length of 4K. For HSA, we set the sliding-window size of SWA heads to 1K and explore sparsity ratios of $\rho = 0.75$. Discrepancy-based KV-head selection follows the same data setup described in Section 4. We compare against layer-wise SWA with four attention sinks (Xiao et al., 2024), using identical training configurations, and keep the first and last layers as full attention. For short-context evaluation, we report results on MMLU (Hendrycks et al., 2021), ARC-Easy/Challenge (Clark et al., 2018), BoolQ (Clark et al., 2019), HellaSwag (Zellers et al., 2019), OpenBookQA (Mihaylov et al., 2018), PIQA (Bisk et al., 2020), and WinoGrande (Sakaguchi et al., 2021). For long-context evaluation, we assess performance on LongBench (Bai et al., 2024). All evaluations are conducted using the Language Model Evaluation Harness (Gao et al., 2024).

**Results.** We present the results in Tables E and F. On short-context benchmarks, all methods perform similarly to the baseline. On long-context evaluations, the limitations of layer-wise sparsity become more apparent, as dropping global context in sparse layers leads to cumulative degradation. In contrast, HSA alleviates this issue by ensuring global information is preserved in every layer, yielding clear advantages at high sparsity ratios. For example, at $\rho = 0.75$, HSA improves performance on LongBench by 1.1% compared to layer-wise SWA.

Table E: Performance of OLMo 2 7B across different methods on short-context benchmarks.

| Method | MMLU | ARC-c | ARC-e | BoolQ | HellaSwag | OpenBookQA | PIQA | WinoGrande | Avg. |
|---|---|---|---|---|---|---|---|---|---|
| Baseline | 60.2 | 57.5 | 82.5 | 79.9 | 80.3 | 45.8 | 81.1 | 74.0 | 70.1 |
| Layer-wise SWA ($\rho = 0.75$) | **60.7** | 55.9 | **82.9** | 79.2 | 80.0 | **46.4** | 80.9 | 73.4 | **69.9** |
| HSA ($\rho = 0.75$) | 60.4 | 55.0 | 81.7 | **80.4** | **80.2** | 46.2 | **81.0** | **74.0** | **69.9** |

Table F: Performance of OLMo 2 7B on LongBench across different methods.

| Model | Single-Doc QA | | | | Multi-Doc QA | | | | Summarization | | | | Few-shot Learning | | | | Code Completion | | Avg. |
|---|---|---|---|---|---|---|---|---|---|---|---|---|---|---|---|---|---|---|---|
| | NQA | QQA | MFQA-en | MFQA-zh | HQA | 2WM | Mus | DuR | GvR | QMS | MNs | VCS | TRC | TQA | SSM | LSHT | LCC | RBP | |
| Baseline | 6.2 | 20.7 | 21.2 | 13.0 | 31.3 | 25.9 | 10.9 | 8.8 | 20.0 | 15.6 | 11.5 | 7.9 | 51.6 | 79.3 | 34.6 | 15.3 | 34.2 | 30.6 | 25.2 |
| Layer-wise SWA ($\rho = 0.75$) | 5.9 | 14.1 | 14.6 | 9.6 | **31.2** | 24.9 | **13.1** | 9.2 | 20.9 | **16.5** | **12.3** | 7.8 | 44.5 | **80.2** | 32.3 | 10.2 | **29.8** | 29.7 | 23.3 |
| HSA ($\rho = 0.75$) | **7.9** | **21.1** | **20.6** | **9.9** | 27.2 | **26.8** | 13.0 | **9.3** | **21.1** | 16.4 | 12.2 | **8.2** | **49.7** | 78.9 | **33.8** | **11.9** | 29.4 | **30.3** | **24.4** |

