# OpenReview forum: "HSA: Head-wise Sparse Attention for Efficient and Accurate Long-context Inference"
_ICLR.cc/2026/Conference — ICLR 2026 Conference Withdrawn Submission_

### Official Review · Reviewer_11nb · 2025-10-30

**Soundness:** 2
**Presentation:** 2
**Contribution:** 2
**Rating:** 2
**Confidence:** 4

**Summary:**

This paper proposes Head-wise Sparse Attention (HSA), a hybrid attention mechanism that applies sliding-window attention (SWA) at the KV-head level to reduce computational and memory costs in long-context inference. The authors introduce a discrepancy-based selection strategy to identify which heads should remain global and which can be sparsified, followed by continued training to adapt the model. Experiments on various benchmarks show that HSA outperforms layer-wise sparse baselines, especially in long-context settings.

**Strengths:**

- The paper is well-structured and clearly motivated, with a thorough evaluation across multiple models and benchmarks.
- The idea of applying sparsity at the head level is intuitive and aligns with the observation that attention patterns vary across heads.
- The experimental results demonstrate consistent improvements over layer-wise sparse baselines, particularly in long-context scenarios.

**Weaknesses:**

- The core components of HSA, sliding window attention and head-wise sparsity, have been extensively explored in prior works such as Mistral[1], StreamingLLM[2], MInference[3], XAttention[4], and FlexPrefill[5], making the contribution of this paper incremental.
- The proposed method requires continued training (e.g., 100B–200B tokens) to adapt pre-trained models to the new sparsity pattern. This is computationally expensive and less practical compared to methods like MInference or FlexPrefill, which require no fine-tuning.
- While StreamingLLM has shown that sliding window attention with attention sinks can maintain performance without fine-tuning, this paper does not provide a zero-shot evaluation of HSA. It remains unclear whether the performance gains are due to the architecture or the extensive continued training.
- Although theoretical reductions in computation and KV-cache are provided, the paper lacks empirical measurements of actual inference speed or memory usage. This makes it difficult to assess the real-world efficiency gains of HSA.
- The paper does not include comparisons and discussions with recent sparse methods [2-8], which are more relevant and competitive.

[1] Mistral 7b. 2023

[2] Efficient streaming language models with attention sinks. ICLR 2024.

[3] MInference 1.0: Accelerating pre-filling for long-context LLMs via dynamic sparse attention. NeurIPS, 2024.

[4] Core Context Aware Transformers for Long Context Language Modeling. ICML 2025.

[5] XAttention: Block Sparse Attention with Antidiagonal Scoring. ICML 2025.

[6] FlexPrefill: A Context-Aware Sparse Attention Mechanism for Efficient Long-Sequence Inference. ICLR 2025.

[7] Curse of High Dimensionality Issue in Transformer for Long Context Modeling. ICML 2025

[8] MMInference: Accelerating Pre-filling for Long-Context VLMs via Modality-Aware Permutation Sparse Attention. ICML 2025

**Questions:**

NA

---

### Official Review · Reviewer_g3vr · 2025-10-31

**Soundness:** 2
**Presentation:** 2
**Contribution:** 2
**Rating:** 2
**Confidence:** 5

**Summary:**

This paper proposes HSA (Head-wise Sparse Attention), a hybrid attention mechanism that applies sliding-window attention (SWA) at the KV-head level to reduce computational cost and KV cache size while preserving global context. The authors introduce a discrepancy-based selection strategy to identify which heads should remain global and which can be sparsified, followed by continued training to adapt the model to the new sparsity pattern. Experiments on in-house and public benchmarks show that HSA outperforms layer-wise sparse baselines, especially in long-context settings.

**Strengths:**

- The paper is well-structured and clearly motivated, with a thorough evaluation across multiple benchmarks and model scales.
- The idea of preserving at least one global KV head per layer is intuitive and helps mitigate the "weakest-link" effect observed in layer-wise sparse models.
- The study includes ablation experiments on window size, attention sinks, and selection strategies, offering useful insights into design choices.

**Weaknesses:**

- The novelty of this paper is limited. The core components of HSA (sliding window attention and head-wise sparsity) have been extensively explored in prior works[1-3]. The head-wise sparsity patterns in these works [2,3] are more diverse and refined than the approach presented here.
- Although theoretical reductions in FLOPs and KV cache are analyzed, the paper lacks empirical measurements of actual inference speed or memory usage.
- The motivation for head-wise sparsity is well-aligned with recent trends, but the literature review does not adequately situate HSA within the existing landscape of head-wise sparse attention methods.
- The selection strategy based on output discrepancy is reasonable, but its dependence on a calibration set and the need for continued training limit its plug-and-play potential.
- The experiments only compare against layer-wise sparse models and omit comparisons with recent head-wise sparse methods (e.g., MInference, XAttention, FlexPrefill). This undermines the claim that HSA offers a superior head-wise sparsity approach.

---

[1] Efficient streaming language models with attention sinks. ICLR 2024.
[2] MInference 1.0: Accelerating pre-filling for long-context LLMs via dynamic sparse attention. NeurIPS, 2024.
[3] FlexPrefill: A Context-Aware Sparse Attention Mechanism for Efficient Long-Sequence Inference. ICLR 2025.

**Questions:**

NA

---

### Official Review · Reviewer_hzFn · 2025-10-31

**Soundness:** 2
**Presentation:** 3
**Contribution:** 2
**Rating:** 2
**Confidence:** 4

**Summary:**

This paper introduces Head-wise Sparse Attention (HSA), which applies sliding-window attention selectively at the KV head level to preserve global context while improving efficiency. A discrepancy-based head selection strategy with continued training achieves better long-context performance than layer-wise sparse baselines.

**Strengths:**

1. Proposes head-level sparsity with a discrepancy-based selection method to preserve global context.
2. Demonstrates consistent improvements over layer-wise sparse baselines on long-context benchmarks.
3. The motivation and method are clearly described with supporting visualizations and ablations.

**Weaknesses:**

1. Methodologically, HSA and DuoAttention[1] are quite similar, but the paper offers little analysis of their differences and no direct experimental comparison.
2. Experiments mostly use in-house models and benchmarks, limiting generalizability and reproducibility. Effectiveness and efficiency tests are conducted on different models, making cross-study comparisons less reliable.
3. HSA requires continued training after head selection, which reduces flexibility. The impact of continued training on accuracy is not well quantified, and Table 5 does not clearly show the effect; more ablation is needed.

[1] DuoAttention: Efficient Long-Context LLM Inference with Retrieval and Streaming Heads

**Questions:**

1. Could the authors clarify the differences between HSA and training-free approaches, such as DuoAttention?
2. Can results be provided on a wider range of open-source models to demonstrate generalizability?
3. Can the efficiency analysis be expanded to include prefill and decoding latency as well as memory usage details?
4. What is the quantitative impact of the continued training step on HSA’s accuracy, and can the authors provide ablation results to assess its necessity and cost?

---

### Official Review · Reviewer_Wkh3 · 2025-11-03

**Soundness:** 2
**Presentation:** 2
**Contribution:** 2
**Rating:** 4
**Confidence:** 5

**Summary:**

The authors propose Head-wise Sparse Attention (HSA), which sparsifies KV heads instead of full layers.

**Strengths:**

The paper highlights the advantage of head-wise sparsity over layer-wise sparsity, a trend that is also evidenced by DuoAttention. This indicates growing consensus that fine-grained head-level control is more effective for preserving global context.

Experimental results consistently favor the proposed approach over layer-wise sparsity, especially at aggressive sparsity levels, where layer-wise methods degrade significantly.

**Weaknesses:**

1. As noted earlier, DuoAttention has already introduced the notion of differentiating between retrieval heads and streaming heads, applying sparsity selectively at the head level. In this sense, the current work appears to be an incremental extension rather than a fundamentally new paradigm. While the proposed discrepancy-based head selection is interesting, the paper should more clearly articulate conceptual differences and advantages relative to DuoAttention.

2. The choice of baselines is somewhat limited. The experiments primarily compare against StreamingLLM and self-ablations, but omit several strong and relevant long-context sparse attention baselines such as DuoAttention, MInference, SeerAttention, and xAttention. Without comparisons to these SOTA methods, it is difficult to fully assess the competitiveness and practical impact of the proposed approach. I strongly suggest including these baselines or providing a compelling justification for their absence.

**Questions:**

See all in the previous.

[1] DuoAttention: Efficient Long-Context LLM Inference with Retrieval and Streaming Heads

[2] MInference 1.0: Accelerating Pre-filling for Long-Context LLMs via Dynamic Sparse Attention

[3] SeerAttention: Learning Intrinsic Sparse Attention in Your LLMs

[4] XAttention: Block Sparse Attention with Antidiagonal Scoring

---

### Note · Authors · 2025-12-03

I have read and agree with the venue's withdrawal policy on behalf of myself and my co-authors.